# Prognostic and Predictive Biomarkers in the Era of Immunotherapy for Lung Cancer

**DOI:** 10.3390/ijms24087577

**Published:** 2023-04-20

**Authors:** Lucile Pabst, Sébastien Lopes, Basil Bertrand, Quentin Creusot, Maria Kotovskaya, Erwan Pencreach, Michèle Beau-Faller, Céline Mascaux

**Affiliations:** 1Pulmonology Department, University Hospital of Strasbourg, 67000 Strasbourg, France; 2Pharmacy Department, University Hospital of Strasbourg, 67000 Strasbourg, France; 3Laboratory Streinth (STress REsponse and INnovative THerapy against Cancer), Inserm UMR_S 1113, IRFAC, Université de Strasbourg, ITI InnoVec, 67000 Strasbourg, France; 4Laboratory of Biochemistry and Molecular Biology, University Hospital of Strasbourg, 67000 Strasbourg, France

**Keywords:** immunotherapy, predictive biomarkers, prognostic, immune checkpoint inhibitors

## Abstract

The therapeutic algorithm of lung cancer has recently been revolutionized by the emergence of immune checkpoint inhibitors. However, an objective and durable response rate remains low with those recent therapies and some patients even experience severe adverse events. Prognostic and predictive biomarkers are therefore needed in order to select patients who will respond. Nowadays, the only validated biomarker is the PD-L1 expression, but its predictive value remains imperfect, and it does not offer any certainty of a sustained response to treatment. With recent progresses in molecular biology, genome sequencing techniques, and the understanding of the immune microenvironment of the tumor and its host, new molecular features have been highlighted. There are evidence in favor of the positive predictive value of the tumor mutational burden, as an example. From the expression of molecular interactions within tumor cells to biomarkers circulating in peripheral blood, many markers have been identified as associated with the response to immunotherapy. In this review, we would like to summarize the latest knowledge about predictive and prognostic biomarkers of immune checkpoint inhibitors efficacy in order to go further in the field of precision immuno-oncology.

## 1. Introduction

Immunotherapy is considered as one of the major advances in the treatment of non-small cell lung cancer (NSCLC). The receptor programmed death-1 (PD-1) is particularly expressed on the surface of CD4/CD8 T cells and dendritic cells (DC) and is involved in the inhibitory signaling of the effector phase of antitumor immunity. Recently, immunotherapy has been shown to be effective in the treatment of cancer patients including those with melanoma [1], NSCLC [2,3], and renal cancer [4]. Immune checkpoint inhibitors (ICIs), such as monoclonal antibodies directed against PD-1 and its ligand programmed death-ligand 1 (PD-L1) are used to block the inhibition of the host cytotoxic T-cell activity against a tumor (Figure 1).

These inhibitors, including nivolumab (anti PD-1), pembrolizumab (anti PD-1), and atezolizumab (anti PD-L1), have demonstrated superiority over second-line chemotherapy after a failure of platinum-based chemotherapy in NSCLC [2,5,6]. Pembrolizumab later became the new first-line standard for patients with NSCLC with high PD-L1 expression (≥50%), including those with or without KRAS mutation [7]. More recently, the combination of chemotherapy and immunotherapy has replaced chemotherapy alone for squamous cell NSCLC [8] and non-squamous cell NSCLC [9] regardless of PD-L1 status.

However, not all patients respond to immunotherapy, encouraging research for predictive biomarkers. In this article, we discuss different types of predictive biomarkers of response to immunotherapy.

## 2. The Role of the Immune Microenvironment in Cancer

The tumor microenvironment (TME) includes the various cell types, vascular structures, signaling molecules, and extracellular matrix surrounding tumor cells. It is a complex setting, hindering immune cell functionality and survivability through features interwoven and overlapping in effect. For simplicity, we discuss here the traits of a prototypical immunosuppressive TME, considered from the cellular, signaling, and metabolic standpoints.

This prototypical TME would abound with stromal and immune cells that are either dysfunctional or polarized into anti-inflammatory phenotypes. Type 2, alternatively activated tumor-associated macrophages (M2 TAMs) are instrumental in orchestrating this immunosuppressive environment. Upon migration from the bloodstream, monocytes receive signals from cancer cells and cancer-associated fibroblasts and differentiate into type 2 TAMs rather than type 1 TAMs, their proinflammatory counterpart. M2 TAMs inhibit immune effector cells and recruit other immunosuppressive cells in the TME. They further support tumor progression by inducing angiogenesis and remodeling the extracellular matrix (ECM) [10,11].

Among other immunosuppressive cells found in the TME are tumor-associated fibroblasts, myeloid-derived suppressor cells (MDSCs), alternatively activated natural killer (NK) cells and dendritic cells, and regulatory lymphocytes. It is now well-established that the contribution of the innate immune system to tumor clearance is as equally vital as that of its adaptive counterpart. Normal NK cells are useful in clearing those tumor cells that do not express MHC class I molecules and escape killing from lymphocytes. Indeed, NK cells rely on a dual system of activating and inhibiting receptors that engage ligands on the surface of the target, and the detection of MHC class I molecules usually acts as an inhibitory signal of degranulation. Tumor-infiltrating NK cells, however, display an altered functionality with downregulated activator receptors and a reduced cytotoxicity [12]. Their secretion profile shifts towards immunosuppressive cytokines (VEGF, IL-10, TGF-β), which are described later.

At the crossroads between innate and adaptive immunity are the dendritic cells (DCs). DCs rely on the detection of danger-associated molecular patterns (DAMPs), as well as a robust inflammatory signal from innate immune cells to pick up and process antigens, migrate to lymph nodes, and trigger an adaptive response from T cells. Oftentimes, DCs are excluded from the TME or do not mature properly. Indeed, they can be driven to a tolerogenic phenotype and express high levels of immunomodulatory ligands (e.g., PD-L1, CTLA-4) and cytokines. Such tolerogenic DCs (tDCs) drive lymphocytes into a tolerant or anergic state and elicit the differentiation of Foxp3+ regulatory lymphocytes (T-regs) [13,14]. This failure to elicit a proper antigen presentation, in the context of a weak inflammatory background, is the first step by which the whole antitumor immune chain reaction is disrupted.

MDSCs are immature myeloid cells that possess various immunosuppressive functions. They elicit T-cell apoptosis through the secretion of reactive oxygen species and the signaling pathways of adenosine and tryptophane. They stimulate CD4+ T-cell tolerance and differentiation into T-regs, which in turn participate in the recruitment of MDSCs [15].

At worst, lymphocytes may be completely absent from tumor beds. Such tumors are said to be noninflamed, or immune deserts, and have a worse prognostic than inflamed tumors. Conversely, many subsets of lymphocytes may be found in the TME, including CD4+ and CD8+ T cells and T-regs. Lymphocytes may be found infiltrating the tumor core or confined to the stromal banks; the latter are said to be excluded [16]. CD4 and CD8 lymphocytes may display various degrees of dysfunction, which culminate in so-called exhaustion. Exhaustion is triggered by a persistent stimulation of TCRs following the failure to clear the antigen load, and its pertinence in physiology lies in the prevention of chronic inflammation. Exhausted lymphocytes display reduced effector functions and an increased expression of inhibitory receptors [17]. Since exhaustion cannot be reversed by immunotherapy, it explains why some tumors, although infiltrated with lymphocytes, do not respond to the treatment [18].

T-regs are either generated by tDCs in lymph nodes or result from the polarization of effector T cells in the TME, and they play a somewhat contrasted role. On the one hand, they are usually associated with cancer aggressiveness and a shorter survival of patients [19]. On the other hand, T-regs have been linked to the slower progression of certain cancer types, including NSCLCs treated with immune checkpoint inhibitors, and a longer patient survival. It is argued that the surge of interferon that accompanies a successful immunotherapy triggers a so-called fragile phenotype in T-regs, turning them into an interferon-producing population, supportive of cytotoxic lymphocytes [20].

There is a reciprocal interaction between cancer cells and connective tissue cells (e.g., fibroblasts, endothelial cells, adipocytes) in the TME. Following the solicitation from cancer cells, stromal cells cause a dysfunction in a way that is supportive of cancer cells and detrimental to proper immune function. Cancer-associated fibroblasts support tumor invasion and metastasis through the production and remodeling of the extracellular matrix (ECM). They signal in a way that inhibits immune effector cells and recruit other immunosuppressive cells in the TME. Cancer-associated adipocytes behave similarly, likewise providing energetic fuels to tumor cells. Endothelial dysfunction leads to impaired an diapedesis and anergy of lymphocytes, through the abnormal expression of inhibitory ligands by endothelial cells [13].

Immunosuppressive signals in the TME notably include the overexpression of immune checkpoints, as well as a secretion imbalance towards anti-inflammatory cytokines. All the aforementioned cell types, including cancer cells, secrete a wide range of cytokines, chemokines, and prostaglandins, of which three are chiefly related to a decreased immune function: IL-10, VEGF, and TGF-β. IL-10 is a long-known anti-inflammatory cytokine. VEGF was discovered through its stimulating effect on tumor vessels, which earned it its name; it also inhibits the function of T cells, polarizes DCs into tDCs, and recruits M2 TAMS and MDSCs [21]. In the immunosuppressive context of the TME, TGF-β adds to the effects of VEGF. It is particularly known to induce the expression of transcription factor Foxp3 in CD4+ T cells, turning them into T-regs, which in turn produce more TGF-β. TGF-β also controls the tumor progression by promoting epithelial-mesenchymal transition (EMT), a state of modified mobility and adherence that enables cell migration and metastasis [22]. Finally, TGF-β leads to the overexpression of indoleamine 2,3-dioxygenase 1 (IDO1) in immune cells. By converting tryptophan into kynurenine, IDO1 has potent modulatory effects, including the suppression of T cells and NK cells and the activation of regulatory T cells and myeloid-derived suppressor cells [23].

Immune checkpoints are important in the establishment of peripheral tolerance and intervene at different stages of lymphocyte maturation. For instance, PD-1 inhibits effector functions of activated lymphocytes, while CTLA-4 counteracts costimulatory signals at the priming stage, and VISTA helps enforcing quiescence in naïve lymphocytes. These, in addition to others (e.g., TIGIT, TIM-3, LAG-3) are hijacked by cancer cells to escape the killing from CD8 T cells [24]. However, the effect of immune checkpoints is not limited to lymphocytes, and they participate to the overall immunosuppressive pressure that TAMs, tDCs, T-regs. and MDSCs exert on immune cells in the TME [25].

Since the neovasculature in tumors is aberrant and tumor cells are overly competitive for nutrients, the TME is a hypoxic, acidic, low-glucose environment, in which it is difficult for other cell types to strive. Hypoxia triggers complex, adaptive changes in tumor cells. These include increased resistance to metabolic stress and oncologic treatments; tumor invasion and metastasis through the promotion of angiogenesis; the promotion of EMT and remodeling of the extracellular matrix by CAFs; and the induction of an immunosuppressive secretion profile [26]. Hypoxia also triggers the release of extracellular adenosine-triphosphate, which is then processed into cyclic monophosphate adenosine by enzymes CD39 and CD73. Adenosine has membrane receptors on immune cells, notably T cells, and acts as an immunomodulatory signal [27].

## 3. Microenvironment and Its Impact on Outcome with Immunotherapy

### 3.1. The Role of PD-L1

Not all NSCLC patients benefit from these immunotherapies. Indeed, the response rate in second-line therapy is between 14% and 20%, depending on the type of molecule [2,6,28]. In first-line therapy, after the selection of patients expressing PD-L1 at different thresholds, response rates are higher (from 26 to 45%) [28]. There is a difference in response depending on whether PD-L1 is expressed or not and also in proportion to the level of PD-L1 expression [9].

PD-L1 expression thus seems to be a predictive marker of response to immunotherapy but the analysis of this factor is not sufficient; indeed, responses are seen in tumors not expressing PD-L1 in 6.5% to 10% of the cases (Table 1) [2,3,6,28,29,30,31,32]. Pooled results at 5 years of phase III trials Checkmate 017 (squamous NSCLC) and 057 (nonsquamous NSCLC) showed a larger benefit of nivolumab compared to docetaxel in patient with PD-L1-positive tumors (HR 0.61, IC 0.49–0.76) and a smaller benefit in patients with PD-L1-negative tumors (HR 0.76, IC 0.61–0.96), remaining statistically significant in both cases [33].

The expression of PD-L1 does not offer any predictive certainty of either a response or resistance to treatment. Moreover, the interpretation can be made difficult by the expression of PD-L1 by the tumor cells themselves but also by immune cells (T lymphocytes, antigen-presenting cells, etc.) located in the immediate vicinity. Finally, tumor heterogeneity may be responsible for false negatives.

### 3.2. The Role of Tumor-Infiltrating Lymphocytes

Tumor-infiltrating lymphocytes or TILs are the subject of much research. The interactions between tumor cells and the different components of the immune system are increasingly known and it is recognized that these interactions are key in the anticancer activity of the immune system as well as in the mechanisms explaining the resistance of some tumors to immunotherapy. Thus, the analysis of the nature of the tumor infiltration could, in the future, at the time of diagnosis, also have a predictive value for the efficacy of immunotherapy treatment and thus guide it.

Thus, many of the previous studies attempted to clarify the importance of the tumor microenvironment and its immune cells in the immunotherapy efficacy [34]. Among these cells are the regulatory T cells, a subgroup of CD4+ TLs that we already mentioned, and which play a critical part in suppressing the effector T-cell response by secreting interleukin 10, interleukin 35, and TGF-β [35,36]. While other CD4+ TLs promote the local immune response, regulatory T cells (T-regs) maintain the immune homeostasis. T-regs cells are particularly expressed in NSCLC tissues and their level is assumed to be associated with the risk of relapse [37]. The CD8 marker is a specific cluster expressed on the surface of T-reg (regulatory) T cells. These are immune cells strongly involved in the activation of cytotoxic T cells, which are important effectors of the tumor response (by the direct destruction of tumor cells).

In particular, three phenotypes describing the intratumor immune response have been described: “inflammatory”, with infiltrations of CD4 and CD8 lymphocytes and an overexpression of the checkpoint inhibitor PD-L1 on tumor cells; “excluded”, with the presence of immune cells but limited to the stroma and not penetrating the tumor; “immune desert”, with the absence of T cells [38].

The immunosuppressive tenant of adaptive immunity is represented by regulatory T cells, which are involved, like immune checkpoints, in the physiology of peripheral tolerance. Tumor cells are known to recruit regulatory T cells from their environment, a molecular marker of which is the transcription factor forkhead box protein 3 (FOXP3) [39,40]. The level of the transcription factor FoxP3 is an indirect measure of the level of T-regs [41]. Studies have therefore evaluated the role of FoxP3 in the regulation of T-regs activity, showing that patients with a defect in FoxP3 expression had an increased probability of autoimmune and inflammatory diseases [42]. It has now been demonstrated that, both in general oncology [43] and specifically for NSCLC [44], tumors with a high infiltration of FoxP3 and T-regs are associated with a poor prognosis in terms of OS. The role of FoxP3 expression in relation to other T-regs subtypes (CD4+, CD25, CD127, etc.) and CD8+ now remains to be clarified [45].

### 3.3. HLA and CMH-1

The HLA-1 (human leukocyte antigen class I) consists of genes coding for the MCH-1 molecules involved in the presentation of intracellular antigenic peptides to naive TCD8 lymphocytes allowing a clonal amplification and cellular activation phase. The HLA-1 varies between individuals due to polymorphisms. Variations are mainly localized on the antigenic peptide-binding region, so that each variant binds a specific repertoire of peptide ligands. Consequently, a homozygous subject in at least one HLA-1 locus would have a smaller and less diverse repertoire of tumor antigen recognition [46], and it has a deleterious impact on the overall survival of these patients [47]. More specifically, Chowell et al. showed an association between HLA gene subtypes and patients’ outcome when treated by ICIs. The HLA B44 supertype was associated with improved survival while the HLA B62 supertype (particularly the 15:01 allele) was associated with lower survival [47]. Mutations in beta-2-microglobulin (B2M) belong to the CMH-1 and is required for the presentation of antigens to the dendritic cells. Thus, mutations in B2M, altering the antigenic presentation, are leading to resistance to ICIs. Among 29.4% of progressions associated with a resistance to ICIs have been shown to be linked to a B2M abnormality [34].

### 3.4. Interferon Signatures

Interferon (IFN) are important molecules implicated in antitumor immune response. Recent publications have shown that interferon signature seems to be a predictive biomarker of ICIs’ response. For example, a high expression of the IFNγ gene was associated with a complete or partial response to ICIs in melanoma patients [48]. In the POPLAR trial, a high IFNγ signature was associated with an improved OS in NSCLC patients treated with atezolizumab [29]. Moreover, IFNs are one of the markers of the tumor inflammation signature (TIS) [49,50]. In fact, PD-L1 expression is closely related to the genetic expression of IFNγ [29,48]. After IFNγR binding of type II interferon, JAK1 and 2 get phosphorylated. STAT1 attach to the receptor and are phosphorylated too. These dimers induce IRF1 and act as a transcription factor in the nucleus of the tumor cell. This pathway regulates the production of PD-L1, which induces its surface expression (Figure 2). This process is one of the hypothesis of the good response to ICIs in a high-interferon signature [51].

### 3.5. The Neutrophil/Lymphocyte Ratio

Because lymphocyte-mediated cytotoxicity leads to the release of cytokines which can inhibit the growth and progression of cancer cells, the presence of lymphocytes within and around cancer cells is associated with a better response to cytotoxic treatment [52]. Neutrophils also play an important role in tumor inflammation [53] by producing chemokines and cytokines that inhibit the immune activity of lymphocytes. A large number of neutrophils in the tumor environment should therefore generate an inflammation response, resulting in the proliferation and metastasis of cancer cells [54]. On the other hand, by being involved in hemostasis, thrombosis, and inflammation, platelets have the ability to support cancer stem cells, induce angiogenesis, and favor metastasis by evading immune detection [55].

In order to find a cost-effective prognostic biomarker for cancer patients, biochemical markers of inflammation (leucocytes and their subtypes, levels of some cytokines, CRP, albumin) have been incorporated in prognostic factors for numerous types of cancer [56]. A prognostic scoring system, the Glasgow Prognostic Score (GPS), based on those acute phase proteins, emerged and has been validated in a variety of cancers, independently of tumor characteristics. The neutrophil-to-lymphocyte ratio (NLR) is another one, based on the hematological components of systemic inflammatory response.

Because this ratio was first presented as a simple tool to measure the intensity of stress and systemic inflammation in critically ill patients [57] and because high counts of neutrophils in blood had already been confirmed as an independent predictive factor in cancer patients [58], Walsh et al. decided to evaluate its prognostic value in colorectal cancer patients and concluded that a preoperative NLR greater than five was correlated with a poorer survival [59]. It was also demonstrated in metastatic melanoma patients that an NLR > 5 was an independent biomarker of a poor prognosis, whatever the treatment received [60]. In an NSCLC adjuvant setting, it has then been shown that the NLR was associated with a higher stage and could be used as a predictive biomarker of survival [61].

With the rise of immunotherapy, the NLR and derived neutrophil-to-lymphocyte ratio (dNLR; absolute neutrophil count/(white blood cell concentration — absolute neutrophil count)) have been investigated as predictive biomarkers for ICIs [62,63]. Those studies therefore confirmed that an inflammatory environment was associated with poor survival in melanoma patients treated with ICI therapy. Different large systematic reviews then confirmed the prognostic value of the NLR in a variety of cancers and regardless of its operability and the treatment received [64,65]. Finally, the normal values of the NLR have been placed between 0.78 and 3.53 [66].

The question of the prognostic value of the NLR in lung cancer soon arose and Bagley et al. showed that a pretreatment NLR ≥ 5 was correlated with an inferior overall survival in NSCLC patients treated with nivolumab [67], with no conclusion regarding its prognostic nor predictive value. A low preoperative NLR was also correlated with a better survival in stage I resected NSCLC [68]. More precisely, a preoperative NLR ≥ 2.5 in stage I NSCLC was shown to lead to a statistically significantly shorter 5-year overall survival [69]. For patients with locally advanced NSCLC (stage IIIA and IIIB), a low pretreatment NLR seems correlated with better survival [70,71]. The same results were observed among stage IV NSCLC patients and the variation of the NLR during the first cycle of treatment could also predict survival [72]. The prognostic impact of dNLR’s early variation under ICI therapy for advanced NSCLC patients was similarly described by Mezquita et al. in another recent publication [73], concluding that a high dNLR that changed to low was associated with a better outcome with immunotherapy.

As a result of all those previous studies, a prognostic index, based on the combination of a baseline dNLR and the LDH level, the Lung Immune Prognostic Index (LIPI), was developed in order to assess the risk of resistance to ICI treatment in patients with advanced NSCLC [74,75]. Based on the LIPI, Mezquita et al. therefore stratified the NSCLC population into three groups: good (low LIPI), intermediate, and poor (high LIPI)

Poor-LIPI patients had a shorter PFS and OS than patients with an intermediate or good LIPI. However, it has to be noted that the LIPI was only correlated with outcome in patient treated with ICIs.

To conclude, accessible, reproductive, and less expensive than other biomarkers, NLR appears to be a promising prognostic biomarker of response to ICI therapy [76]. Combining it with other biomarkers could make it even more effective. The platelet-to-lymphocyte ratio (PLR), as an example, has been reported as another prognostic factor for patients’ various cancers [65].

## 4. The Role of Mutations

The tumor mutational burden (TMB) corresponds to the number of somatic mutations per megabase of DNA. These mutations lead to the production of neoantigens that can be recognized by the immune system and the associated antitumor response. It could therefore be a predictive factor for ICI’s effectiveness when the mutational load is high (≥10 mut/Mb) [35,36,77] with nonsynonymous mutations [78,79]. This cutoff is not absolute and could vary depending on the molecular technique used (whole genome sequencing, WGS; whole exome sequencing, WES; targeted next-generation sequencing, NGS) as well as the different panel of genes. In cholangiocarcinoma, patients with a high mutational load treated with anti-PD-1 were able to achieve complete remission [80].

Another cause of tumor antigen enhancement is based on the instability of genome repeat sequences or microsatellites. This instability is a consequence of the inactivation of the DNA mismatch repair (MMR) system. These are called MSI (microsatellite instability) tumors. Considered as an interesting predictive biomarker, the FDA has approved pembrolizumab for unresectable solid MSI-H (high microsatellite instability) tumors [41,42,43]. Furthermore, MSI-H tumors tend to have a high mutational burden [43]. Nevertheless, contrary to colorectal cancer, MSI in NSCLC is a very rare event.

POLE and POLD1 encode the ε and δ catalytic subunits of DNA polymerase involved in DNA replication and repair. Thus, the function of POLE and POLD1 is essential to suppress gene mutations and consequently tumor genesis [45,81]. POLE/POLD1 mutant tumors are recognized by the immune system, making the tumor immunogenic. In several cancers, these mutations have shown to be an interesting predictive factor. For example, patients with POLE-mutated endometrial cancer had a better response to treatment [82,83]. This association was also found in non-small cell lung cancer [84,85]. Nevertheless, POLE/POLD1 mutations appear not so frequent in NSCLC.

If a high mutational burden is associated with good response to ICIs, some oncogenic addictions are factors of poor prognosis under immunotherapy. Indeed, in the majority of patients with EGFR-mutated lung cancers, the response to anti-PD-1 is compromised. The mechanism of resistance would be explained by a decrease in PD-L1 expression, a low mutational burden, and a reduced lymphocyte infiltration as “cold” tumors [86]. EGFR mutations have also been associated with a hyperprogression under ICIs as well as MDM2/4 amplifications or mutations of tumor suppressor genes such as TSC2 and VHL [87,88]. ALK rearrangement also seems to be a primary resistance factor to ICIs since the ImmunoTarget study showed an objective response rate of 0%, a control of 32%, and a median progression-free survival of 2.5 months with immunotherapy alone [89]. Similar results were observed in the ATLANTIC study [90]. Other mutations searched by NGS such as MET exon 14, HER2, RET rearrangement are also markers of primary resistance to immunotherapy. In contrast, the data are less clear for BRAF V600E and non-V600E mutations [91].

Beside somatic mutations with oncogenic addiction, other somatic mutations detected by NGS could play a role in the sensitivity to ICIs in NSCLC. STK11/LKB1 is a tumor suppressor gene [92] encoding the LKB1 protein. STK11/LKB1 mutations are detected in 10–20% of NSCLC [93] and could result in a “cold” tumor with poor responses to ICIs [94,95].

KEAP1 mutations could also be a negative predictive biomarker for ICIs. Indeed, KEAP1 mutated tumors have a lower infiltration of TCD8 lymphocytes [96]. KEAP1 mutations are associated with higher levels of TILs and therefore should logically be associated with better responses to immunotherapy. However, it has been shown that despite an increased level of TILs, in case of KEAP1/STK11 comutation, there seems to be a primary resistance to ICIs [97]. This was shown in one of the KEYNOTE-042 subgroup analyses comparing pembrolizumab to chemotherapy [98].

This resistance is even more marked in the case of STK11/KRAS comutation, in which PDL1 and TILs expressions are low, whereas KRAS/TP53 comutation seems to be associated with a certain sensitivity to immunotherapy [99].

## 5. Intracellular Signaling Pathways

The WNT protein binds to the N-terminal extracellular domain of a frizzled family receptor, therefore disrupting the destruction complex of β-catenin (by inactivating the GSK-3β activity), which leads to the cytoplasmic accumulation of β-catenin and its nuclear translocation [100]. This pathway is known as the canonical WNT–β-catenin signaling (Figure 3). The noncanonical pathway involves planar cell polarity and WNT–Ca^2+^ pathways [101]. In the absence of WNT ligand, the intracellular concentration of β-catenin stays low thanks to the β-catenin destruction complex.

The WNT/β-catenin pathway plays an essential role in the development of lymphocytes and regulates homeostasis, stem cell control, and wound repair. This pathway is involved in the self-renewal capacity of nonmalignant stem cells, and its aberration would therefore promote the stem-cell-like qualities of tumor stem cells. Alteration in the WNT pathway leads to changes in the tumor microenvironment especially by regulating the activation and differentiation of T cells.

Melanoma was the first cancer in which somatic alteration of the WNT/β-catenin pathway was shown to be associated with the non-T-cell-inflamed tumor microenvironment [102]. Spranger et al. therefore identified, using mouse melanoma models, the mechanism by which tumor-intrinsic active β-catenin signaling led to T-cell exclusion and consequently to a resistance to anti-PD-L1/anti-CTLA-4 therapy [103]. It has now been demonstrated that a deregulated WNT/β-catenin pathway can be found in all stages of oncogenesis, from malignant transformation to metastatic dissemination and resistance to treatment [104,105].

In teratoma models, an enhanced expression of WNT was demonstrated to be correlated with impaired T- and B-cell infiltration [106]. The WNT/β-catenin pathway can also be activated without mutation, with an increased expression of WNT ligands or an increased expression of Frizzled receptors as an example. Resistance to ICIs can also be explained by the fact that the WNT/β-catenin pathway interacts with other proteins that will be developed later as well. Indeed, an active WNT pathway disrupts Foxp3 transcriptional activity which is essential for the development and good functioning of T-regs cells [107]. Higher β-catenin cytoplasmic concentrations have also been associated with enhanced T-reg infiltration and activity [108].

In NSCLC, the activation of the WNT/β-catenin pathway is correlated with a higher TMB [109,110] and a low expression of PD-L1 [111]. There seems to be an inverse correlation between the level of β-catenin and the level of TILs and CD11c+ infiltration [110]. In different cancers, a reduced activation rate of CD8+-TLs has been shown in patients with a high β-catenin expression [112,113,114]. This expression of β-catenin seems to occur under ICIs and would therefore be in favor of an acquired resistance mechanism [115]. Targeting the WNT/β-catenin signaling pathway in combination with ICI therapy could be a way to restore T-cell infiltration and thus maintain sensitivity to immunotherapy [116].

Similarly, the MAPK/PTEN/PI3K pathway is believed to be involved in the resistance to immunotherapy. The loss of PTEN and the activation of PI3K reduce the level of cytotoxic T cells in the tumor, which promotes the resistance to ICIs [117].

## 6. The Role of Microbiota

The response level to ICIs seems to be closely correlated with the composition of the gut microbiota [118]. Indeed, several bacterial species have been found in abundance in patients responding to ICIs while others were seen more abundant in nonresponder patients. Research began with mice models, which showed an increase of *Bifidobacterium breve*, *B. adolescentis,* and also *B. longum* in responders to ICIs [119]. Studies about the impact of microbiota on ICIs’ outcomes are ongoing worldwide. Human studies are shown in Table 2.

Conflicting results were found with *Bacteroides thetaiotaomicron,* as this bacterium was present in both responders and nonresponders [120,121]. To date, no conclusion can be reached with these bacteria. *Akkermansia muciniphila* seems to be positively correlated with a response to ICIs. Routy et al. showed that oral supplementation of this bacterium alone and/or with *Enterococcus hirae* restored ICI response in mice [122]. However, the role of *A. muciniphila* is more complex than at first glance. Indeed, a recent study showed that the best survival response was obtained when *A. muciniphila* was present in small quantities (median survival of 27.2 months compared with 7.8 months when it was present in large quantities and 15.5 months when it was absent from the digestive tract) [123]. This bacterium seems to be a good predictive biomarker candidate for immunotherapy and would possibly have therapeutic interest in nonresponders. More recently, a Japanese team attempted to supplement a cohort of NSCLC patients with *Clostridium butyrium* [124]. This bacterium improved the efficacy of ICIs in the supplemented group. This link between microbiota and ICIs’ response is a burning topic and numerous clinical trials to assess bacterium supplementation or fecal microbiota transplantation (FMT) and response outcomes are underway (e.g., NCT04521075, NCT04577729, and NCT04988841 for FMT, NCT03637803, NCT03775850, and NCT04601402 for probiotics supplementation).

We already know that gut microbiota modulates adaptive and innate immunity and thus influences anticancer immune response in the tumor microenvironment [125] by immunostimulatory effects and other complex mechanisms. Several bacteria present in responder patients have shown different types of immune modulations. As an example, *B. fragilis* activates Th1 polarized lymphocytes and presents a cross-reactivity between bacterial and tumor antigens [126]. *Bifidobacteria* improve IFN-γ production by TCD8 lymphocytes and tumor infiltration, while *Akkermansia muciniphila* enhances IL-12 production by dendritic cells [122]. Another hypothesis was the activation of intratumoral and splenic dendritic cells (DC) by *bifidobacterium,* which improved tumor-specific CD8+ T-cell responses (2). Microbial or pathogen-associated molecular patterns (MAMPs or PAMPs) from the microbiota are also involved, such as lipopolysaccharide (LPS), which enhances T cell activity [127]. Indeed, bacterial DNA modulates the balance between T-regs and effector T cells [128]. Microbial metabolites may be involved as well in the immune system modulation [129]. For example, short-chain fatty acids modify cytokine production and dendritic cell activity [130,131]. Inosine, a purine metabolite, produced by *A. muciniphila* or species of Bifidobacterium play an important role in the response to ICIs [132]. This metabolite improves tumor immunogenicity and enhances the capacity of the antitumor cell to present antigen [133]. The IFNɣ pathway seems to be increased in tumor cells when in presence of inosine, which can activate NK cells and their cytotoxic activity. The TNFα pathway can be increased as well. which can enhance antigen presentation [133]. Inosine seems to enhance the immune cells’ activation by adenosine A2 receptor, which promotes T1-cell differentiation in the population growth in the tumor microenvironment [132]. Gut microbiota may also have an impact on B-cell class switching, enabling T-regs cells differentiation [134,135].

In the past few years, drug–drug interaction between ICIs, antibiotics (ATB), and proton-pump inhibitors (PPI) have been raised. The majority of publications are retrospective studies, but a meta-analysis seems to demonstrate a negative impact of this medication on ICIs’ outcomes [136,137]. The decrease in overall survival and progression-free survival by ATB and PPI would probably result from the balance between the gut microbiota and immune checkpoint inhibitors.

**Table 2 ijms-24-07577-t002:** Human studies which assess the link between ICIs’ response and gut microbiota.

Studies	Location	Number of Patients	Cancer Type	ICI Type	Analysis	Findings
Chaput N. et al., 2019 [138]	Europe	38	Melanoma	Anti-CTLA-4	16rRNA gene sequencing	*Faecalibacterium* and *firmicutes*: better response to ICIs; *bacteroides*: poor response
Frankel AE. et al., 2017 [139]	America	39	Melanoma	Anti-CTLA-4, anti-PD-1	16rRNA gene sequencing and metagenomic shotgun sequencing	*Bacteroides caccae*, *Faecalibacterium prausnitzii*, *Bacteroides thetaiotaomicron*, *holdemania filiformis*, *dorea formicognerans*: good responders
Fukuoka S. et al., 2018 [140]	Asia	38	NSCLC and gastric cancer	Anti-PD-1	16rRNA gene sequencing	high alpha diversity, *Ruminococcaceae*: ICI responders
Gopalakrishnan V. et al., 2017 [120]	America	89	Melanoma	Anti-PD-1	16rRNA gene sequencing	high alpha diversity, *Clostridium*, *Ruminococcaceae*: enriched in responders; *Bacteroides thetaiotaomicron*, *Escherichia coli*, low alpha diversity: poor responders
Jin Y. et al., 2019 [141]	Asia	42	NSCLC	Anti-PD-1	16rRNA gene sequencing	*Alistipes*, *Bifidobacterium longum*, *Parvotela copri* and high alpha diversity: better response; unclassified *Ruminococcus*: enriched in nonresponders
Maia M. et al., 2018 [142]	America	16	RCC	Anti-PD-1	16rRNA gene sequencing	*Roseburia* and *Faecalibacterium* spp.: ICI responders
Matson V. et al., 2018 [143]	America	42	Melanoma	Anti-PD-1 and anti-CTLA-4	Metagenomic shotgun sequencing, 16rRNA gene sequencing, and polymerase chain reaction	*Klebsiella pneumoniae*, *Veillonella parvula*, *Parabacteroides merdae*, *Lactobacillus* sp.: response to ICIs; *Ruminococcus obeum*, *Roseburia intestinalis*: poorer response
Peters B. et al., 2019 [144]	America	27	Melanoma	Anti-PD-1 and anti-CTLA-4	Metagenomic shotgun sequencing and 16rRNA gene sequencing	*Faecalibacterium prausnitzii*, *Coprococcus eutactus*, *Prevotella stercorea*, *Streptococcus sanguinis*, *Streptococcus anginosus*, and *Lachnospiraceae bacterium* 3 1 46FAA: longer PFS; *Bacteroides ovatus*, *Bacteroides dorei*, *Bacteroides massiliensis*, *Ruminococcus gnavus*, and *Blautia producta*: shorter PFS
Routy B. et al., 2018 [122]	Europe	100	NSCLC and RCC	PD-1 and anti-PD-L1	Metagenomic shotgun sequencing	Akkermansia muciniphila, *Alistipes*, *Eubacterium*, *Ruminococcus*: better response to ICIs; *Parabacteroides distasonis*: poor responders
Vetizou M. et al., 2015 [121]	Europe	25	Melanoma	Anti-CTLA-4	16rRNA gene sequencing	*Bacteroides fragilis* and *Bacteroides thetaiotaomicron*: good responders
Zheng Y. et al., 2019 [145]	Asia	8	Melanoma	Anti-PD-1	Metagenomic shotgun sequencing	High alpha diversity: good response to ICIs

RNA: ribonucleic acid, ICI: immune checkpoint inhibitor, PFS: progression-free survival, NSCLC: non-small cell lung cancer, RCC: renal cell carcinoma.

## 7. The Role of Radiomics

Currently, the new area for reflection concerns radiomic biomarkers, which consist of using artificial intelligence (AI) algorithms to automatically quantify radiographic features. This method is built on the assumption that imaging data are the result of mechanisms located at a genetic and molecular level, related to the genotypic and phenotypic characteristics of the tissues, and therefore, that they can be used as a prognostic biomarker [66,67,68,69,70]. Using scannographic characteristics as biomarkers has several advantages. First, this noninvasive procedure is performed routinely in patients with cancer. Second, it allows the visualization and assessment of the whole tumor as well as any sites of metastatic disease and provides thus a better assessment of any heterogeneity that exists within or between tumors, driving differential prognoses and responses to treatment. Consequently, it offers opportunities to escalate or change treatment, or consider multimodality treatment options, such as the addition of stereotactic ablative body radiotherapy (SABR) for an oligoprogressive disease.

A study published by Trebeschi et al. in 2019 showed that, if the group of patients with a homogeneous response to treatment was from far the group with the best prognosis, among patients with a progressive tumor, those with both responsive and progressive lesions had a better prognosis that those with a uniform progression [146]. The same authors showed that radiomics better predicted patients’ response and progressions overall, and the superiority of radiomics for predicting the response was particularly seen in lung- and lymph-node lesions of NSCLC. In Liu et al.’s study, including training and independent test sets of 112 and 49 patients, respectively, the radiomic signature at baseline had no significant predictive value regarding response status. A combined Delta-radiomics nomogram with clinical factor lesions was able to distinguish responders from nonresponders (AUCs of 0.83 (95% CI: 0.75–0.91) and 0.81 (95% CI: 0.68–0.95) in the training and test sets, respectively). In a subset of those patients with available pretreatment PD-L1 expression status (n = 66), models incorporating Delta-radiomic features showed a superior predictive accuracy compared to that of the PD-L1 expression status alone (*p* < 0.001). Notably, if 285 patients were screened for eligibility for this trial, only 161 could be analyzed in the radiomic cohort, illustrating that such a signature is not applicable to all patients due to various factors including the quality of the CT scan or the lack of target lesion [147]. Mu et al. analyzed data from 194 patients (99 in a retrospective training set, 47 in a retrospective test set, and 48 in a prospective test set). They identified a radiomic signature of F-FDG PET/CT scans predicting patients with durable clinical benefit with AUCs of 0.86 (95% CI: 0.79–0.94) in the training set, 0.83 (95% CI: 0.71–0.94) in the validation retrospective set, and 0.81 (95% CI: 0.68–0.92) in the retrospective test cohort [148]. A new class of quantitative radiomic biomarkers of tumor-associated vasculature (QuanTAV) appears to hold great promise. A team from Cleveland presented their ability to predict outcomes by comparing different treatment regimens and combining observations of vascular morphology with CT and MRI [149].

Based on 15 published studies, a meta-analysis was performed to assess the quality of radiomic studies for predicting immunotherapy response and outcome in patients with non-small cell lung cancer. The pooled diagnostic odds ratio for predicting immunotherapy response in NSCLC was 14.99 (8.66–25.95). High- and low-risk groups were identified both for OS (pooled HR: 1.96, 95% CI 1.61–2.40, *p* < 0.001) and for PFS (pooled HR: 2.39, 95% CI 1.69–3.38, *p* < 0.001) [150]. Although the results of this meta-analysis confirmed the interest of radiomics for predicting the response and outcome of NSCLC to treatment by immunotherapy, it remains difficult to predict how this methodology will be standardized for use in routine practice. Indeed, so far, each individual study has been monocentric and used a homemade radiomic signature.

## 8. Immunogenic Cell Death

The combination of chemotherapy and immunotherapy has been the standard treatment for NSCLC for some years. The efficacy of this combination is not simply a matter of adding the cytotoxic mechanisms of chemotherapy and immunotherapy. Several studies suggest that chemotherapy induces immunogenic cell death (ICD), which potentiates the effects of immunotherapy [104].

ICD is thought to result in the release of immunostimulatory molecules from tumor cells, leading to the activation of antigenic presentation and thus of the adaptive immune response. Markers of ICD include damage-Associated molecular patterns (DAMPs), which include, among others, ATP release, calreticulin membrane resettlement (CRT), or HMGB1 expression.

One pathway that appears to be strongly involved in DCI is the type I interferon pathway, which is involved in the secretion of cytokines and immunomodulatory chemokines (CXCL9, CXCL10) that bind to the CXCR3 receptor and in the recruitment of CD8+ T cells to the tumor microenvironment [151]. A recent study in murine-model bronchial carcinomas resistant to combinations of chemotherapy and immunotherapy showed that chemotherapy did not increase the expression of CXCL9 and 10. The INF type 1 pathway appears to be required for DCI [152].

One of the reference chemotherapy molecules that induces DCI is oxaliplatin (14), a platinum salt used in digestive cancers, but this drug is not used in the treatment of NSCLC. Pemetrexed is a chemotherapy of the antifolate family that is commonly used in the treatment of bronchial adenocarcinoma. It was shown that after treatment with this molecule, the tumor immune environment was mainly composed of CD8+ (cytotoxic) lymphocytes. This immune stimulation is explained by an increase in the expression of genes involved in immune cell cytotoxicity (PRF1, GMZA, and GZMB) and in the maturation of immune cells through the expression of certain endoplasmic reticulum stress molecules such as ATP and TLR4 (HMGB1) as well as an IFN 1 type response (CXCL 10). This immune stimulation effect is also found with paclitaxel, a taxane chemotherapy used in the treatment of NSCLC [153]. For carboplatin and cisplatin, which are the most used platinum salts in NSCLC, the results seem to be more discordant, and some studies even show a decrease in DCI induced by pemetrexed or paclitaxel when combined with platinum salts [153].

Furthermore, it would appear that the doses of chemotherapy required to induce DCI are lower than those inducing cytotoxic effects.

In a study, MEK inhibitors such as trametinib were shown to restore the induction of CXCL9 and 10 when treated with the platinum salt and pemetrexed doublet, resulting in a reduced tumor progression and increased survival. The activation of the MEK pathway is thought to inhibit mitophagy by modulating optineurin (OTP). However, mitophagy appears to be necessary for the production of CXCL 9 and 10 due to the activation of TLR9 by mitochondrial DNA in the cytoplasm [152].

Chemotherapies such as platinum salts and pemetrexed are known to damage mitochondria [154], whereas MEK inhibitors are thought to allow the reinduction of mitophagy in damaged mitochondria. The destruction of mitochondria leads to the release of mitochondrial DNA which, by associating with TLR9, activates the IFN1 response and stimulates the production of CXCL9 and 10. These chemokines induce the recruitment of CD8+ T cells into the tumor environment and activate their cytotoxic function. Note that blocking CXCL9 by using antibodies against CXCL9 does not seem to have an impact on ICD, so CXCL 10 seems to be the key factor in DCI [152].

## 9. Conclusions

If immune checkpoint inhibitors targeting PD-1 and PD-L1 have radically changed the NSCLC treatment landscape and its prognosis, only twenty percent of patients will finally really benefit from it, and identifying those patients seems necessary in order to prevent potential toxicity. Nowadays, PD-L1 tissue testing is the only validated marker used to select potential ICIs responders.

However, PD-L1 expression as the only predictive biomarker has some limitations: tissue biopsy is not always feasible, and even on tissue biopsy, there can be a spatial or temporal heterogeneity; moreover, PD-L1 expression can be modified and altered by previous treatment lines, and even with a high PD-L1 expression status, some patients will not respond to ICIs and may even suffer immune-related adverse events while PD-L1-negative tumors might have durable responses to ICIs. 

The need for additional predictive and prognostic biomarkers in lung cancer has never been more urgent. In this review, we discussed different biomarkers related to the efficacy of ICIs and their potential prognostic value in lung cancer (Table 3), including TMB, TILs, IFNγ, NLR, and the composition of the gut microbiota [155]. The association of different biomarkers such as PD-L1 expression and microsatellite instability/mismatch repair deficiency could also serve as an indicator of ICI potential responders [156]. However, even with commonly used biomarkers such as PD-L1 and TMB, there is no unified detection standard with different conclusions based on the same biomarkers. Research today aims to identify “biosignatures”, i.e., genetic, transcriptomic, proteomic, and radiomic profiles of markers, which can reflect the multitude of pathogenic mechanisms.

Nevertheless, some biomarkers discussed in this review are contradictory and require further prospective studies in order to validate the most promising predictive and prognostic biomarkers. Composite biomarkers incorporating different variables should also be implemented to maximize their predictive power. Once the predictors of response to immunotherapy have been validated, the next step will be to select those that can be most easily analyzed. Indeed, with the rise of liquid biopsy for diagnostic purposes, the next step would be an equivalent in peripheral blood for predictive or prognostic intents. Recently, many circulating-tumor-DNA-based technologies have been developed, and if they are still mainly used as a complement to tissue genotyping in order to detect genomic alterations [157], they can also be used as a real-time assessment of therapeutic response, using plasma-based changes in ctDNA levels. The predictive value of ctDNA is being investigated since ctDNA clearance has been described as a predictive biomarker for patients treated with immunotherapy. Free-ctDNA fragmentomics have already shown a certain potential to predict postoperative relapse risk, therefore selecting patients for whom adjuvant therapy would improve recurrence-free survival [158]. For locally advanced patients, positive ctDNA tests after chemo-radiotherapy have been shown to be a predictive marker of clinical benefit with ICIs. Repetitive samplings during the course of therapy are therefore emerging as a noninvasive method to guide immunotherapy in selected patients with advanced and metastatic solid tumors [159]. By detecting ctDNA mutations and assessing bTMB and MSI, liquid biopsy could help selecting patients for ICI.

Moreover, with the possibility of ICI resistance among patients during treatment, studies should not only focus on the prognostic and predictive values of biomarkers for ICIs but also on their potential role in the mechanisms related to ICI resistance.

Finally, with the emergence of ICI combinations for the treatment of NSCLC, the question of the place of the previously mentioned biomarkers will soon arise, ultimately prompting research towards a precision immuno-oncology.

## Figures and Tables

**Figure 1 ijms-24-07577-f001:**
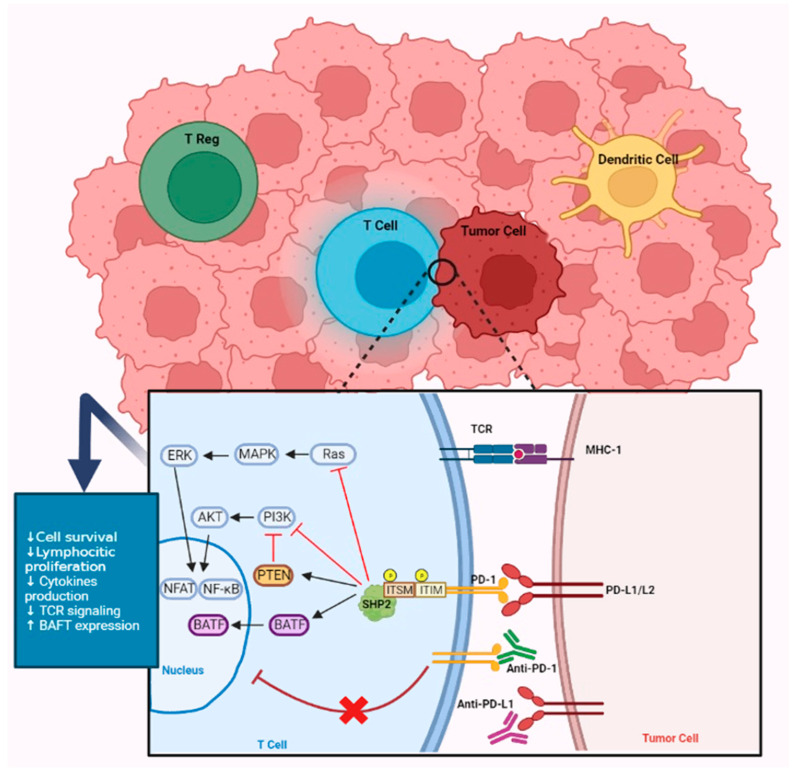
PD1/PD-L1 pathway. PD-1/PD-L1 interaction leads to phosphorylation of ITIM and ITSIM patterns, resulting in the recruitment of a phosphatase (SHP-2). SHP-2 will therefore dephosphorylate several key kinases in the TCR signaling pathway, in particular Ras and phosphoinositide 3-kinase (PI3K). It results in a decrease of transcription factors such as the nuclear factor of activated T cells (NFAT) and nuclear factor-κB (NF-κB), which play an important role in LT activation, proliferation, survival, and effector functions. Furthermore, LT functions can be inhibited by an increased expression of BATF (basic leucine zipper ATF-like), a transcription factor. Antibodies against PD-1 and PD-L1 stop the inhibitor signal and restore T-cell function against tumor.

**Figure 2 ijms-24-07577-f002:**
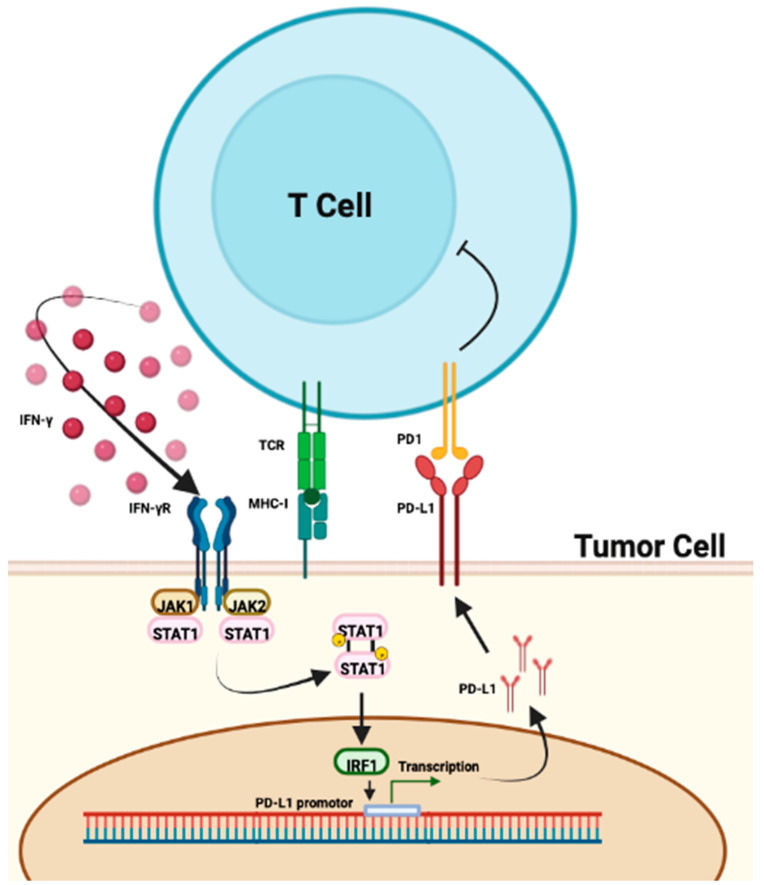
Pathway of PD-L1 regulation by IFNγ. PD-L1 expression by tumor cells correlates with IFNγ production by T cells. Binding of interferon to its receptor induces phosphorylation of JAK1 and 2 and then STAT1. STAT1 dimers induce IRF1 which leads to transcription of the PD-L1 promoter. This pathway regulates the production of PD-L1 and its expression on the surface of tumor cells.

**Figure 3 ijms-24-07577-f003:**
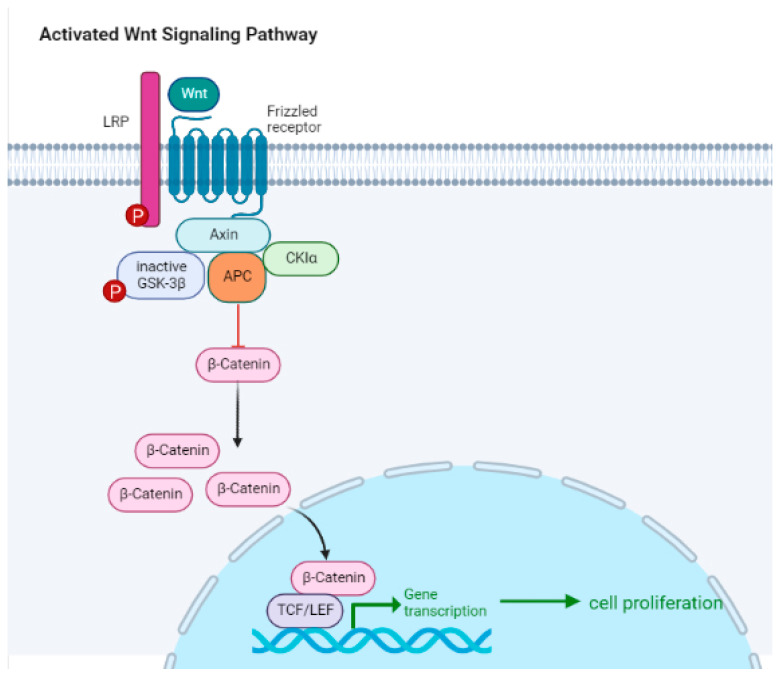
The canonical WNT/β-catenin pathway. In this pathway, the WNT ligand binds to a receptor complex (a frizzled receptor associated to a LRP co-receptor) on the surface of the cell, activating a series of downstream signaling events by disrupting the tertiary complex formed by Axin, adenomatous polyposis coli (APC), CK1α, and GSK-3β, that ultimately result in the accumulation of β-catenin in the nucleus. Once in the nucleus, β-catenin interacts with transcription factors (TCF-Lef transcriptional complexes) to promote the expression of genes involved in cell proliferation, differentiation, and survival. However, in the absence of WNT signaling, β-catenin is rapidly degraded by the activated tertiary complex formed by GSK-3β, APC, and Axin. Thus, the balance between active WNT signaling and β-catenin degradation is critical for proper cell function and homeostasis.

**Table 1 ijms-24-07577-t001:** Efficacy of immune checkpoint inhibitors against PD-1 or PD-L1 in PD-L1-negative tumors.

Drug	Study Name	Histology	Testing	Cut-off PD-L1	% PD-L1	ORR
Nivolumab	Checkmate 017 [33]	Squamous	Dako 28.8	<1%	40%	17%
Nivolumab	Checkmate 057 [33]	Nonsquamous	Dako 28.8	<1%	46%	9%
Atezolizumab	Poplar [29]	All histologies	Ventana SP142	TC0 + IC0	32%	7.8%
Atezolizumab	Oak [7]	All histologies	Ventana SP142	TC0 + IC0	45%	8%
Durvalumab	(Phase I–II) [30,31]	All histologies	Ventana SP263	<25%	45%	6.1%
Pembrolizumab	(Phase I) [6]	All histologies	Dako 22C3	<1%	39%	8.1%
Avelumab	(Phase Ib) [32]	All histologies	Dako 73.10	<1%	14%	10%

**Table 3 ijms-24-07577-t003:** Main available described biomarkers for immunotherapy and their limitations.

Biomarker	Type	Limitations
PD-L1 expression	Predictive	Limited specificity and sensitivity, variations between tumor types and sites; can also vary over time and treatment course
Tumor mutational burden and mutations	Predictive	No standardized measurement method as of today and may vary between tumor types and sites
Tumor-infiltrating lymphocytes	Prognostic	Limited standardization in TILs’ quantification
LIPI (Lung Immune Prognostic Index)	Prognostic	Limited to lung cancer
NLR (neutrophil-to-lymphocyte ratio)	Prognostic	No standardization and may be influenced by multiple factors such as inflammation and infection
Microsatellite instability	Predictive	Limited to some kinds of cancers only
Intracellular signaling pathways	Predictive	Still a limited understanding of the different known pathways and their interaction with the immune system; may vary between cancer types
Gut microbiota	Predictive	May be influenced by multiple factors such as diet, antibiotics use, proton-pump inhibitors (PPI) use, and other medications
Radiomics	Predictive/prognostic	Limited standardization and validation; may vary with imaging techniques

## Data Availability

Not applicable.

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
