# Peer review of "Prognostic and Predictive Biomarkers in the Era of Immunotherapy for Lung Cancer"

_ijms, 2023, doi:10.3390/ijms24087577_

Round 1

Reviewer 1 Report

The manuscript is well written and covers important aspects of immunotherapy in lung cancer.

I recommend the authors include a small section discussing the contribution of fragmentomics approach in the described setting.

The manuscript would also benefit from a short section on authors' point of view as to the specific markers/techniques that will drive the advancement in the field (not only generally described as in conclusions). AI tools/applications should also be elaborated in more detail in this section.

Author Response

Dear Reviewer;

Thank you for your review. We added as requested a section on circulating free tumor DNA profiling as a predictive and patient-monitoring biomarker.

Kind regards,

Dr L. Pabst

Reviewer 2 Report

Reviewer comments

Lucile Pabst and the group discussed the prognostic and predictive biomarkers in the era of immunotherapy in this article. According to the article, the available biomarkers are not sufficient and we urgently need additional predictive and prognostic biomarkers in lung cancer. The title is general for immunotherapy however main text including the introduction part is discussing lung cancer. Either mention the lung cancer in the title or add more data on other cancer too.

Overall, the paper is interesting, but the structure is confused. The structure of the paper has to be improved; the discussion and presentation of results have to be clarified before being reconsidered for publication using suggested comments.

A major revision is required.

Major comment

1.      The title is “Prognostic and predictive biomarkers in the era of immunotherapy” however complete MS has discussed only lung cancer.

2.      Introduction is very short and has two paragraphs only and both paragraphs have discussed only lung cancer.

3.      Add more specific content regarding prognostic and predictive biomarkers in the introduction section with proper citations to enhance the quality of the paper.

4.      In section 2, paragraph 5 is mentioned (described later) however in the later section there is no discussion about the signaling pathways of tryptophane.

5.      Need to incorporate a summary table for available prognostic & predictive biomarkers with their limitations.

6.      Citation is written in the wrong format. Citations should be provided exactly in the journal’s specific format. Please check the citation style on the IJMS journal website.

7.      In conclusion, paragraph 3 mentioned “Therefore, the need for additional predictive and prognostic biomarkers in lung cancer..…..” and the last paragraph “Finally, with the emergence of ICIs combinations for the treatment of NSCLC….” is further focusing on lung cancer only.

Minor comments

1.      Write the full form with abbreviations for t-reg for the first time writing in MS and check the writing pattern for T-regs. (In some places T-regs while on other Tregs)

2.      In abstract lines 4 and 9, make a space after a full stop.

3.      Table 1 is insufficient to provide the data. Need to add more data to it.

4.      Table 1, does not have any citations. Add proper citations in the table.

5.      Give a space after B. in “B.adolescentis and also B.longum”.

6.      In the last paragraph of section 7 you have written “Based on 15 published studied” however in you have cited only 1 reference i.e. reference no 119.

7.      Reference the following articles in the discussion section

https://doi.org/10.1038/s41585-022-00676-0

https://doi.org/10.1186/s40364-020-00209-0

https://doi.org/10.3389/fonc.2021.617335.

Author Response

Dear reviewer;

Thank you for your inputs. Here are the modifications we made : 

Major comments :

  1. We changed the title of this review to "Prognostic and predictive biomarkers in the era of immunotherapy for lung cancer"
  2. see modification n°1
  3. In order to please another reviewer comment we added specific citations later in the review but therefore did not modify the introduction otherwise there would have been too much repetition.
  4. We removed the (described later).
  5. We made a third table with the different main biomarkers and their respective limitations in the conclusion section.
  6. We changed the citation model to the IJMS model.
  7. see modification n°1

Minor comments : 

  1. We added the "T regulatory cells" explanation and changed all "Tregs" to "T-regs".
  2. We verified that every full stop was followed by a space.
  3. and 4. We added the citations asked in Table 1.
  4. see n°3
  5.  We added the requested space.
  6. The n° 119 reference corresponds to the meta analysis which contains the 15 published studies, therefore there is only one citation indeed.
  7. We added the 2nd and third citations but, focusing only on lung cancer, we could not add the first one.

Hoping to have been able to meet your expectations with the modifications made.

Kind regards,

Dr L. Pabst

Reviewer 3 Report

The review “Prognostic and predictive biomarkers in the era of immunotherapy” by Pabst et. al offers a detailed and neatly summarized information regarding immunotherapies, primarily the Immune checkpoint inhibitors in treating cancers. The review also improves the therapeutic understanding regarding the monoclonal antibodies directed against Programmed Death (PD-1) and its ligand (PD-L1).

The review by Pabst and group is of high clinical relevance and value. The review also discusses different features of an immunosuppressive tumor microenvironment. It’s very interesting to read relevance of microbiota in modulating response to immune checkpoint inhibitors.

Figures and table provided are clear and easy to understand.

The review can be accepted in its current form.

Author Response

Dear Reviewer;

Thank you for your review.

Round 2

Reviewer 2 Report

1. The authors have included all suggested points and done all required corrections in the revised MS. 

2. Reference style is not exactly followed in all. Please check once more for the reference style.

3. After corrections in the format of the reference, the MS could be accepted.

Author Response

Dear reviewer;

We changed the format of the references and made some changes (there were some mistakes with the references associated with the part on mutations).

Thank you again for your input;

Best regards,

Dr L. Pabst